# Inferring effects of mutations on SARS-CoV-2 transmission from genomic surveillance data

Brian Lee [1], Ahmed Abdul Quadeer [2,3], Muhammad Saqib Sohail[2,4], Elizabeth Finney[1], Syed Faraz Ahmed[2,3,5], Matthew R. McKay [2,3,5,6] ✉ & John P. Barton [1,7,8] ✉

New and more transmissible variants of SARS-CoV-2 have arisen multiple times over the course of the pandemic. Rapidly identifying mutations that affect transmission could improve our understanding of viral biology and highlight new variants that warrant further study. Here we develop a generic, analytical epidemiological model to infer the transmission effects of mutations from genomic surveillance data. Applying our model to SARS-CoV-2 data across many regions, we find multiple mutations that substantially affect the transmission rate, both within and outside the Spike protein. The mutations that we infer to have the largest effects on transmission are strongly supported by experimental evidence from prior studies. Importantly, our model detects lineages with increased transmission even at low frequencies. As an example, we infer significant transmission advantages for the Alpha, Delta, and Omicron variants shortly after their appearances in regional data, when they comprised only around 1-2% of sample sequences. Our model thus facilitates the rapid identification of variants and mutations that affect transmission from genomic surveillance data.

Viruses can acquire mutations that affect how efficiently they infect new hosts, for example by increasing viral load or escaping host immunity[1–4]. The ability to rapidly identify mutations that increase transmission could inform outbreak control efforts and identify potential immune escape variants[5–9]. However, estimating how individual mutations affect viral transmission is a challenging problem.

To address this challenge, we developed a method to infer the effects of single nucleotide variants (SNVs) on viral transmission that systematically integrates genomic surveillance data from different regions. Our analytical approach is based on a simple epidemiological model, allowing it to be efficiently applied to large data sets and opening the door to future theoretical extensions. Our method is also

automatic in the sense that it relies only on sequence data and does not require, for example, clustering sequences into discrete "variants." An additional advantage of our approach is that relative changes in viral transmission are statistically explained in terms of the specific mutations that different viruses bear, highlighting mutations that may be especially biologically important. Simulations show that our approach can reliably estimate the transmission effects of SNVs even from limited data. As our approach is based on surveillance data, it infers the effects of observed SNVs, rather than predicting the effects of SNVs that have never been observed before.

We applied our method to more than 7.4 million SARS-CoV-2 sequences from 149 geographical regions to reveal the effects of SNVs

[1]Department of Physics and Astronomy, University of California, Riverside, Riverside, CA, USA. [2]Department of Electronic and Computer Engineering, Hong Kong University of Science and Technology, Clear Water Bay, Hong Kong SAR, China. [3]Department of Electrical and Electronic Engineering, University of Melbourne, Melbourne, VIC, Australia. [4]Department of Computer Sciences, Bahria University, Lahore, Punjab, Pakistan. [5]Department of Microbiology and Immunology, University of Melbourne, at The Peter Doherty Institute for Infection and Immunity, Melbourne, VIC, Australia. [6]Victorian Infectious Diseases Reference Laboratory, Royal Melbourne Hospital, Melbourne, VIC, Australia. [7]Department of Physics and Astronomy, University of Pittsburgh, Pittsburgh, PA, USA. [8]Department of Computational and Systems Biology, University of Pittsburgh School of Medicine, Pittsburgh, PA, USA. ✉e-mail: matthew.mckay@unimelb.edu.au; jpbarton@pitt.edu

on viral transmission throughout the pandemic. While the vast majority of SARS-CoV-2 SNVs have negligible effects, we readily observe increased transmission for sets of SNVs in Spike and other hotspots throughout the genome. For clarity, we will refer to all non-reference nucleotides (including deletions or insertions) as SNVs and viral lineages possessing common sets of SNVs as variants. When we discuss the effects of amino acid substitutions, we will use the term mutations to distinguish these substitutions from variation at the nucleotide level, as multiple different nucleotide changes can lead to the same change at the level of proteins.

Importantly, our approach is sensitive enough to identify variants with increased transmission before they reach high frequencies. This is demonstrated by studying the rise of the Alpha and Delta variants in Great Britain and Omicron in South Africa. We reliably infer increased transmission for these variants soon after their emergence, when their frequency in the region was only around 1–2%. An untargeted search for sets of SNVs that strongly increase viral transmission also reveals multiple collections of SNVs belonging to well-known variants. Collectively, these data show that our model can be applied for the surveillance of evolving pathogens to robustly identify variants with transmission advantages and to highlight key SNVs that may be driving changes in transmission.

## Results

### Epidemiological model

To quantify the effects of SNVs on viral transmission, we developed a generalized Galton–Watson-like stochastic branching process model of disease spread (see the "Methods" section). Branching processes have been frequently used to model the stochastic numbers of infections in a population[10,11]. Our model draws the number of secondary infections caused by an infected individual from a negative binomial distribution with mean $R$, referred to as the effective reproduction number, and dispersion parameter $k$. The negative binomial distribution for secondary infections has been used in past work to model superspreading[12–17], which becomes more prominent when $k$ is small. However, we show in the Supplementary Information that the incorporation of detailed information about the current number of infections and degree of superspreading—even if known perfectly—can actually lead to worse inference when data is finitely sampled. Thus, we will ultimately absorb these terms into a regularization parameter, and the degree of superspreading does not influence the inferred transmission effects of SARS-CoV-2 mutations that we will describe below. Multiple variants with different transmission rates are included by assigning a variant $a$ an effective reproduction number $R_a = R(1 + w_a)$. Under an additive model, the net increase or decrease in transmission for a variant is the sum of the individual transmission effects $s_i$ for each SNV $i$ that the variant contains. In analogy with population genetics, we refer to the $w_a$ and $s_i$ as selection coefficients. We will maintain this analogy throughout this work, associating natural selection or fitness with the relative capacity for viral transmission between hosts.

We then apply Bayesian inference to estimate the transmission effects of SNVs that best explain the observed evolutionary history of an outbreak. To simplify our analysis, we use a path integral technique from statistical physics, recently applied in the context of population genetics[18], to efficiently quantify the probability of the model parameters given the data (for details, see Supplementary Information). This allows us to derive an analytical estimate for the maximum a posteriori selection coefficients $\hat{s}$, normalized per serial interval, for a given set of viral genomic surveillance data

$$\hat{s} = \left[ \gamma' I + C_{\text{int}} \right]^{-1} \Delta x . \tag{1}$$

Here $\Delta x$ is the change in the SNV frequency vector over time, $\gamma'$ is a rescaled regularization term proportional to the precision of a Gaussian prior distribution for the selection coefficients $s_i$ (Methods), and $I$

is the identity matrix. The dispersion parameter $k$ and number of infected individuals $N$, analogous to a population size in population genetics, are absorbed into the definition of $\gamma'$. $C_{\text{int}}$ is the covariance matrix of SNV frequencies integrated over time and accounts for competition between variants as well as the speed of growth for different viral lineages (Supplementary Information). Data from multiple outbreaks can be combined by summing contributions to the integrated covariance and frequency change from each individual trajectory (see the "Methods" section). Our theoretical model could also be extended to incorporate additional features of disease transmission, such as the travel of infected individuals between different outbreak regions.

### Validation in simulations

To test our ability to reliably infer the transmission effects of SNVs, we analyzed simulation data using a wide range of parameters. We found that inference is accurate even without abundant data, especially when we combine information from outbreaks in different regions (Fig. 1, Supplementary Fig. 1) Because we model the evolution of relative frequencies of different variants, accurate inference of transmission effects does not require the knowledge of difficult-to-estimate parameters such as the current number of infected individuals or the effective reproduction number (see the "Methods" section). Simulations also demonstrated that our model is robust to variations in effective reproduction numbers in different regions (Supplementary Fig. 2).

### Global patterns of selection in SARS-CoV-2

We studied the evolutionary history of SARS-CoV-2 using genomic data from GISAID[19] as of January 26, 2024. We separated data by region and estimated selection coefficients jointly over all regions (see the "Methods" section). After filtering regions with low or infrequent coverage, our analysis included more than 7.4 million SARS-CoV-2 sequences from 149 different regions, containing 1398 nonsynonymous SNVs observed at nontrivial frequencies.

Our analysis revealed that, while the majority of SNVs were nearly neutral, a few dramatically increased viral transmission (Fig. 2a, Supplementary Table 1). We observe clusters of SNVs with strong effects on transmission along the SARS-CoV-2 genome (Fig. 2b). The highest density of SNVs that increase transmission is in Spike, especially in the S1 subunit (Supplementary Fig. 3). Of the top 20 mutations at the amino acid level that we infer to be most strongly selected, 16 are in Spike (Supplementary Table 1). However, SNVs with a strong selective advantage are also found in other proteins, especially in N, NSP4, NSP6, and NSP12.

### Mutations inferred to strongly increase transmission

The top 50 mutations that we infer to increase SARS-CoV-2 transmission the most are listed in Fig. 2c and Supplementary Table 1. These mutations are at the amino acid level and can include the effect of multiple SNVs and deletions that occur within a single codon. Experimental evidence directly or indirectly supports 48 of these 50 inferences. For clarity, we will reference mutations at the amino acid level rather than the underlying SNVs, which are also given in Supplementary Table 1.

Spike mutations F486P, Q498R, Q954H, N460K, P681R, R346T, N969K, and N679K comprise 8 of the top 10 mutations, and all have demonstrated functional effects that could increase transmission[20,20–25]. Similarly, Spike mutations in the receptor binding motif (RBM) such as F486P, Q498R, N460K, N450D, T478K, N501Y, L452R, and the so-called FLip mutations L455F and F456L appear prominently in our analysis, comprising 9 of the top 25 mutations. Most of these mutations have been shown to increase resistance to RBM-specific neutralizing antibodies[20–22,24,26] and the majority also enhance ACE2 receptor binding[4,21,27–31].

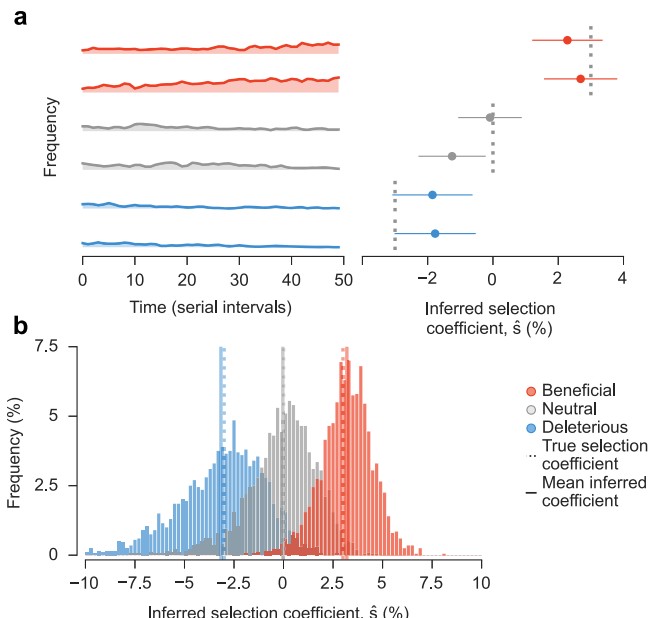

**Fig. 1 | Our approach accurately estimates transmission effects of SNVs in simulations.** Simulated epidemiological dynamics begin with a mixed population containing variants with beneficial, neutral, and deleterious SNVs. **a** Selection coefficients for individual SNVs, shown as mean values ± one theoretical s.d. (i.e., the width of the posterior distribution, see Supplementary Information), can be accurately inferred from stochastic dynamics in a typical simulation. **b** Extensive tests on 1000 replicate simulations with identical parameters show that inferred selection coefficients are centered around their true values. Deleterious coefficients are slightly more challenging to infer accurately due to their low frequencies in data. Simulation parameters. The initial population is a mixture of two variants with beneficial SNVs ($s = 0.03$), two with neutral SNVs ($s = 0$), and two with deleterious SNVs ($s = -0.03$). The number of newly infected individuals per serial interval rises rapidly from 6000 to around 10,000 and stays nearly constant thereafter. Dispersion parameter $k$ is fixed at 0.1.

Of these, N501Y ($\hat{s} = 10.1\%$, ranked 15th) is shared by almost all major SARS-CoV-2 variants. Q498R, N460K, and T478K are shared by all Omicron variants. Beyond the functional effects above, N501Y is known to increase transmission of infection[32] and to help maintain Spike in an active conformation for receptor recognition[21]. Eight Spike N-terminal domain (NTD) mutations/deletions (T19R/I, Δ142, Δ157/F157L, H245N, A264D, and G142D) are also strongly selected. These lie in the antigenic supersite where mutations have been shown to decrease the neutralization potency of NTD-specific monoclonal antibodies[21,33]. Spike mutations unique to the recently emerged Omicron variants BA.2.86 and JN.1 (N450D, V445H, K356T, E554K, H245N, and A246D) along with those found in the KP.2 and KP.3 variants which have become globally dominant in 2024 (R346T, F456L), rank among the top mutations identified in our analysis. All these mutations are known to impact either ACE2 receptor binding or antibody neutralization[26,34–37].

Research on viral transmission has naturally focused on Spike because of its role in viral entry and as a target of neutralizing antibodies. However, our analysis also reveals strongly selected mutations outside of Spike. These include the NSP4 mutation T492I, and Nucleocapsid mutations R203M/K, Δ32/R32C, and P13L. NSP4 mutation T492I ($\hat{s} = 16.6\%$, ranked 2nd) was reported to increase viral replication and infectivity, enhance cleavage of the viral protease NSP5, and contribute to immune evasion based on experiments and animal models[38]. Nucleocapsid mutation R203M ($\hat{s} = 11.4\%$, ranked 13th) is in the linker region of the protein and enhances viral RNA replication, delivery, and packaging, which may increase transmission[39]. Studies suggest that NSP6 mutations Δ106 and S106T

(ranked 3rd and 38th, $\hat{s} = 16.5$ and $\hat{s} = 6.6$) and F108L (ranked 23rd, $\hat{s} = 7.6$) may increase transmission by interferon antagonism[40]. We also find additional mutations outside of Spike, such as G671S in the RNA-dependent RNA polymerase NSP12 and Δ32 in N, that are highly selected and may be good targets for further experimental study. Our model thus highlights non-Spike mutations that may confer a selective advantage to emerging variants.

## Estimates of relative transmission rates for major SARS-CoV-2 variants

We estimated the net increase in viral transmission relative to the WIV04 reference sequence for well-known SARS-CoV-2 variants by adding contributions from the individual variant-defining SNVs (Fig. 3 and Supplementary Fig. 4, see the "Methods" section). Because our model uses global data and infers the transmission effects of individual SNVs, variants can be compared to one another directly even if they arose on different genetic backgrounds, or if they appeared in different regions or at different times. This also allows us to infer substantially increased transmission for variants such as Gamma or Mu, which never achieved the level of global dominance exhibited by variants like Alpha, Delta, Omicron, or XBB (Supplementary Fig. 4).

Our findings are consistent with past estimates that have shown a substantial transmission advantage first for Alpha and then for Delta relative to other pre-Omicron lineages[41–43]. However, past estimates have varied substantially depending on the data source and method of inference. In different analyses, Delta has been inferred to have an advantage of between 34% and 97% relative to other pre-Omicron lineages[41,42,44]. Similarly, Alpha has been estimated to increase transmission by 29–90% relative to pre-existing lineages in different regions[5,41,45–47]. One advantage of our approach is that it can infer selection coefficients that best explain the growth or decline of variants across many regions, allowing for more even comparisons.

Over the period of data that we analyzed, Omicron and its subvariants display clear, large increases in transmission over past variants (Fig. 3). The transmission advantage of BA.1 ($\hat{w} = 170\%$), which we estimate to be the least transmissible of Omicron subvariants, is roughly twice as large as the inferred selection coefficient for Delta ($\hat{w} = 85\%$). More recent variants of Omicron, such as XBB ($\hat{w} = 280\%$) are inferred to be substantially more transmissible.

In general, we find that the contributions of individual SNVs to the overall selection coefficient $\hat{w}$ for a variant are very heterogeneous. A small fraction of SNVs are responsible for most of the increase in transmission. As an example, Supplementary Fig. 5 shows the relative contribution of each Alpha, Delta, and Omicron (BA.1) SNV to the total selection coefficient $\hat{w}$ for the variant. In each case, fewer than 20% of SNVs are responsible for more than 80% of the increase in transmission.

## Detection of low-frequency SNVs that increase transmission

We asked whether strong increases in transmission could be inferred for beneficial SNVs when they are still at low frequencies before they dominate the viral population. To explore this, we considered the rise of the three major variants of concern (VOCs): Alpha and Delta in Great Britain, and Omicron (BA.1) in South Africa. We computed the inferred selection coefficient $\hat{w}$ for each variant in each region at different points in time, as the VOCs began increasing in frequency. Selection coefficients were computed at different times by filtering the sequence data from GISAID to exclude sequences after a specific cutoff date. Note that this approach is different from previous sections, which used all data through January 26, 2024 to compute selection coefficients. To focus on the transmission effects of novel SNVs, we removed putative beneficial SNVs that had been previously observed in other VOCs from the estimates of $\hat{w}$.

We found that the inferred selection coefficients for novel Alpha SNVs rose rapidly as the variant was emerging (Fig. 4a). At the time that

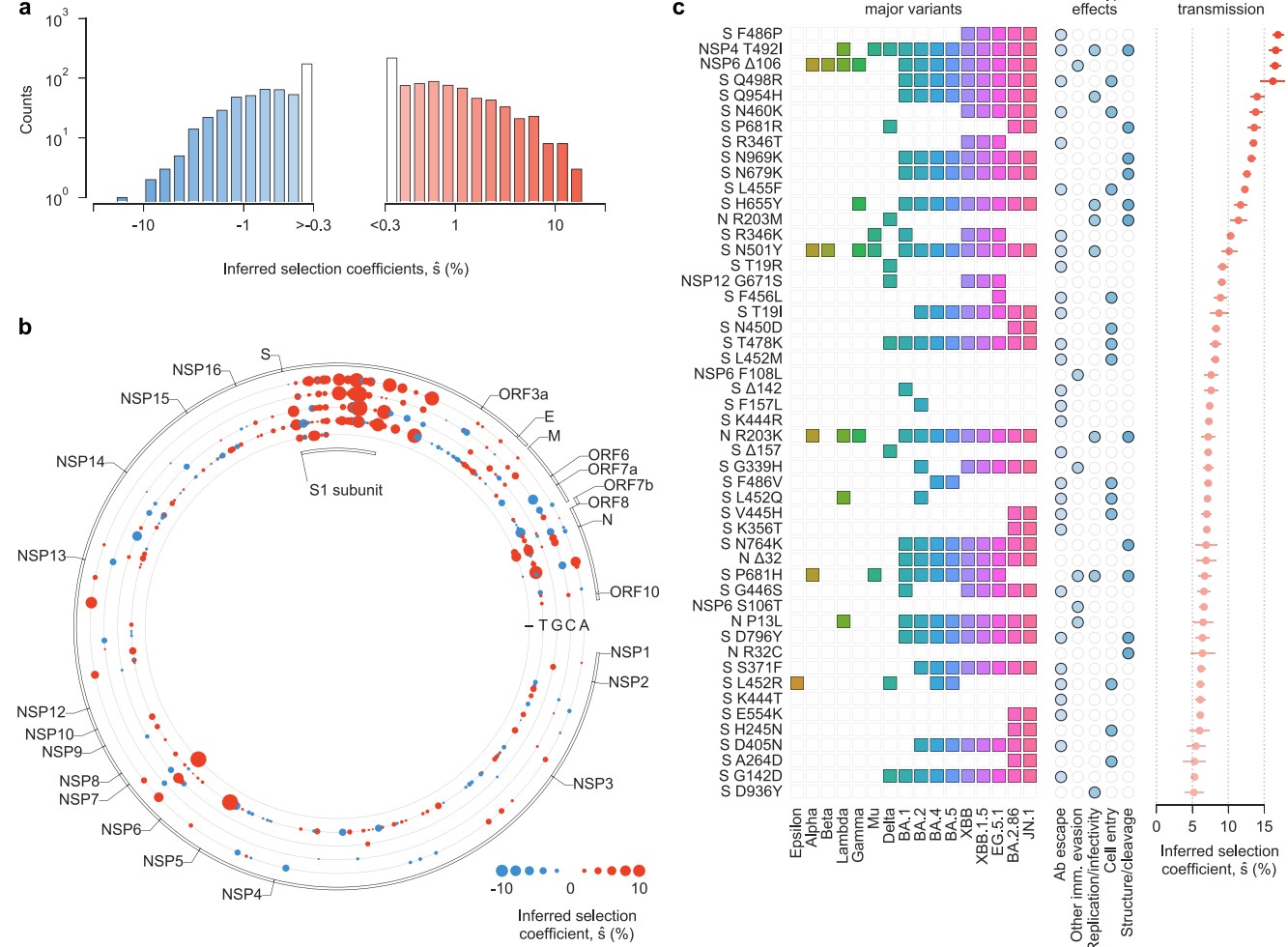

**Fig. 2 | Inferred transmission effects of SARS-CoV-2 SNVs. a** The majority of the 1320 nonsynonymous SNVs included in our study are inferred to have negligible effects on transmission (that is, $\hat{s}$ close to zero). However, a few SNVs have strong effects, as evidenced by a large value of $\hat{s}$. **b** Patterns of transmission effects of SNVs across the SARS-CoV-2 genome. Beneficial SNVs often cluster together in the genome. Clustering is especially apparent for the S1 subunit of Spike, where many SNVs that are inferred to have the largest effects on transmission are located. **c** Top 50 mutations at the amino acid level, which can include the effects of multiple SNVs for deletions and amino acid substitutions that are the result of multiple nucleotide changes within a single codon, inferred to increase SARS-CoV-2 transmission the most, the major variants in which they are observed, their phenotypic effects, and selection coefficients (see Supplementary Table 1). The same colors are used to represent each major variant in Figs. 3, 4 and Supplementary Fig. 4. We cluster experimental phenotypic results into five categories: antibody evasion; other immune evasion; increases in replication and/or infectivity; ACE2 receptor binding and cell entry; and mutations affecting protein structure and/or cleavage.

Public Health England labeled Alpha a variant of interest (VOI)[48], the inferred selection coefficient for novel Alpha SNVs was around 15%. When Alpha was declared a VOC[49], this had grown to around 45%. These statistics would indicate a substantial transmission advantage for Alpha relative to co-circulating variants. Notably, we inferred novel Alpha SNVs to be strongly beneficial even while the variant remained at low frequencies in Great Britain.

Similar analyses show that our model rapidly infers increased transmission for novel SNVs in Delta and Omicron. The selection coefficient for novel Delta SNVs in Great Britain was around 60–70% when it was classified as a VOC[50] (Fig. 4b). No full-length Omicron sequences were available on GISAID when it was designated as a VOC[51]. However, the first Omicron data from South Africa uploaded on December 7, 2021, clearly revealed an enormous transmission advantage for Omicron (Fig. 4c).

In each of these examples, a strong increase in transmission was detectable even for variants at low frequencies. To illustrate this point, we filtered SARS-CoV-2 sequence data by its collection date in each of these regions and computed the frequency of the Alpha, Delta, and Omicron variants over time. At the time that each variant reached a

frequency of 2% in the population, their inferred selection coefficients for novel SNVs were 11%, 16%, and 21% for Alpha, Delta, and Omicron, respectively. These results show that our model can identify SNVs associated with higher transmission even when they are present in a small fraction of all infections in a region.

## Robust identification of beneficial SNVs

Identifying variants that increase transmission in real time could inform public health efforts and highlight important aspects of viral biology. However, the inherent stochasticity of infection and data collection makes accurate inferences difficult. For example, neutral or modestly deleterious SNVs may initially appear to be beneficial due to a transient rise in frequency despite having no selective advantage.

To explore the effects of fluctuations on estimates of transmission effects, we first quantified the inferred selective advantage for all variants $\hat{w}$ (including both SNVs and collections of SNVs that are strongly linked to one another, see the "Methods" section) in each region, for each day that data was submitted to GISAID. As in the previous section, data was filtered by submission date, such that selection coefficients computed for a specific date used only

sequences that were submitted to GISAID on or before that date. Here, we progressively step through time in each region, adding sequences according to their submission date and re-analyzing the data in each region separately.

Although variation in sampling could produce temporary spikes in inferred selection coefficients, we reasoned that large $\hat{w}$ are much more likely to be observed for variants with real, substantial advantages in transmission. To test this reasoning, we used the $\hat{w}$ to identify variants with especially large inferred effects on transmission, which we refer to as high growth (HG) variants ($\hat{w} > \theta$ for some threshold value $\theta$). In each region, we began at the first time point that data was submitted to GISAID and stepped through each subsequent upload date. At each step, we classified strongly linked SNVs with $\hat{w} > \theta$ as HG and excluded these SNVs from future analysis in the same region.

While it is difficult to conclusively determine whether the classification of a group of SNVs as HG is "correct" or "incorrect", we conservatively assumed that (groups of) SNVs in major variants denoted by Greek characters or the B.1 variant should be correctly labeled as HG (true positives), and any other SNVs classified as HG constitute false positives. With this convention, the fraction of true positives increases steadily along with the threshold $\theta$, such that more than 95% of variants classified as HG are true positives for $\theta \geq 18.5\%$ (Supplementary Fig. 6). Thus, variants with inferred selection coefficients $\hat{w} > 18.5\%$ in any region and at any time are highly likely to have a substantial transmission advantage. This threshold could then be used to highlight new variants of particular interest.

We further studied the cumulative fraction of variant-defining SNVs that were classified as HG for 10 major SARS-CoV-2 variants, over time and in 7 broad geographical regions (Supplementary Fig. 7). Despite our stringent threshold of $\theta = 18.5\%$, a large fraction of variant-defining SNVs are ultimately found in HG groups in one or more regions. HG groups encompassing most SARS-CoV-2 variants were also independently detected across different regions, usually within a short period. Importantly, for these variants, around 10–30% of variant-defining SNVs were classified as HG before the variants began wide circulation among humans. This means that not only were some variant-defining SNVs observed in prior variants, but they were also highlighted in our approach as SNVs that were likely to substantially increase SARS-CoV-2 transmission.

### Features of HG SNVs not in major variants

At the threshold value of $\theta = 18.5\%$, we found 38 groups of strongly linked SNVs that did not belong to major, Greek letter variants or B.1. Some of these groups of SNVs may have been identified as HG due to sampling noise. However, others may have biological effects that affect transmission, but not enough to outcompete more transmissible variants. Thus, we investigated whether SNVs in this list could have plausibly affected transmission.

Of the 38 groups, 12 sets of SNVs included Spike mutations at the amino acid level with experimentally demonstrated effects or that lie in functionally important locations. Mutations A879S and A626S were experimentally shown to reduce sensitivity to convalescent sera[26,52].

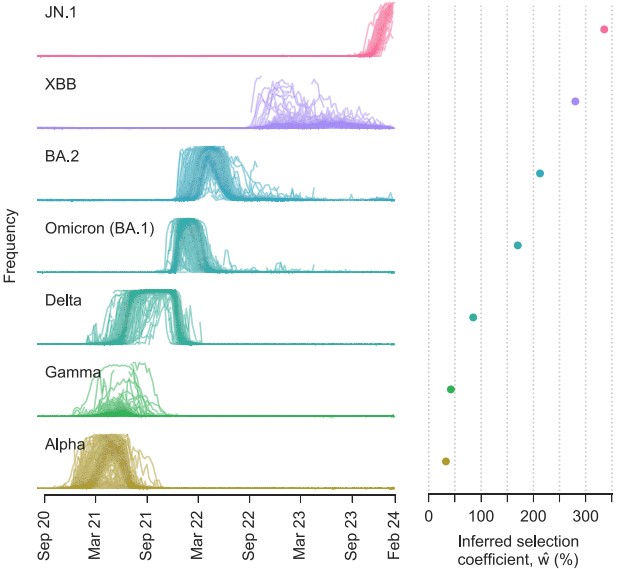

**Fig. 3 | Multiple SARS-CoV-2 variants strongly increase transmission rate.** Frequencies of major variants and their total inferred selection coefficients, shown as mean values ± one s.d. from bootstrap subsampling of regional data (see the "Methods" section), defined relative to the WIV04 reference sequence. Selection coefficients for variants with multiple SNVs are obtained by summing the effects of all variant-defining SNVs.

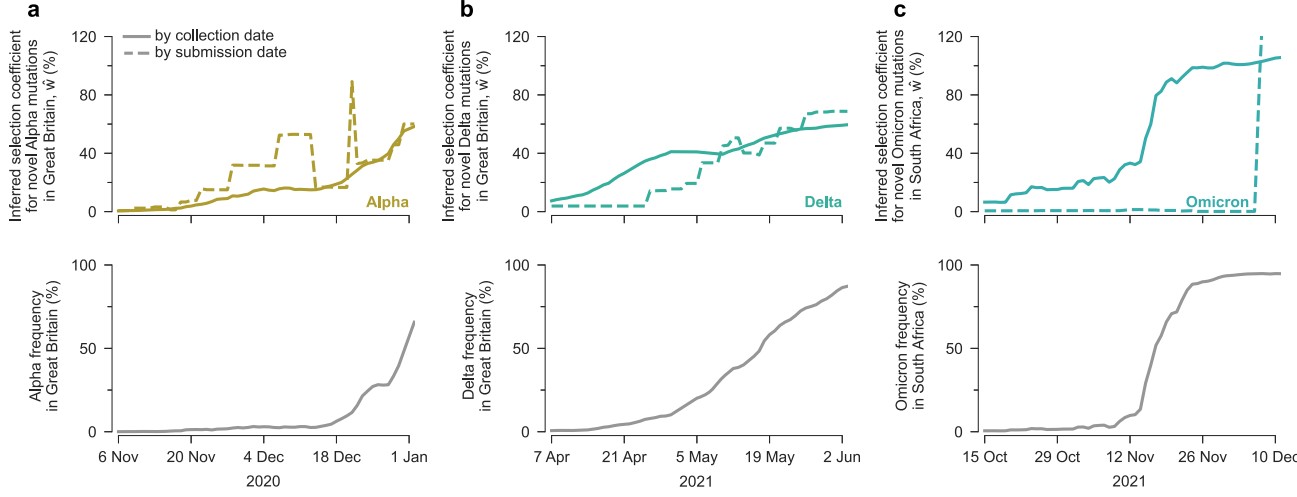

**Fig. 4 | Our model rapidly infers increased transmission for Alpha, Delta, and Omicron (BA.1) SNVs.** Inferred selection coefficients for novel SNVs in Alpha in Great Britain (**a**), Delta in Great Britain (**b**), and Omicron (BA.1) in South Africa (**c**) over time. Selection coefficients were computed over time according to GISAID data filtered by collection date or submission date. Selection coefficients given for a particular date include only data collected or submitted on or before that date. Variant frequencies are computed using sequence data filtered by collection date.

D138Y and W152R/L were shown to escape neutralization by specific antibodies[31,53], and N439K had reduced sensitivity to sera and antibodies[52,54]. N439K and A520S increase binding to the ACE2 receptor[54,55]. In addition, I794N lies on the fusion peptide and on the surface of the Spike protein[56], while Q677P and S680P lie on the furin cleavage site[57,58]. In summary, a substantial fraction of HG Spike SNVs that are not present in major variants could plausibly affect transmission, even if their effects are more modest than some SNVs in major variants.

## Discussion

Quantifying the effects of mutations on viral transmission is an important but challenging problem. We developed a flexible, branching process-based epidemiological model that provides analytical estimates for the transmission effects of SNVs from genomic surveillance data. Applying our model to SARS-CoV-2 data, we identified SNVs that substantially increase viral transmission, including both experimentally validated Spike mutations and other, less-studied mutations that may be promising targets for future investigation. Importantly, we found that our model is sensitive enough to detect substantial transmission advantages for SNVs belonging to major variants even when they comprised only a small fraction of the total number of infections in a region.

Distinct from our method, current approaches to estimate changes in viral transmission often rely on phylogenetic analyses or fitting changes in variant frequencies to logistic or multinomial growth models[5,46,47,59–61]. Phylogenetic analyses for viruses can be challenging due to a high degree of sequence similarity, which implies that the data can be explained equally well by a number of different trees[62], and they also typically rely on Markov chain Monte Carlo sampling that becomes intractable for large data sets. Growth models have been commonly applied to predict relative growth of SARS-CoV-2 variants, and have been incorporated into the popular NextStrain tool[63]. These models can estimate the difference in transmissibility between one variant and others circulating in the same region. However, their estimates may be difficult to compare for variants that arose in other regions or with different genetic backgrounds, and they typically do not identify specific SNVs responsible for changes in transmission.

Our approach differs from these due to our focus on explaining transmission differences between variants by the fitness contributions of individual SNVs. The scalable, analytical form of our estimator for fitness effects also allows for the natural integration of data from multiple regions. The predictions of our model are strongly supported by biological and experimental data. Phenotypic effects have been established for nearly all (i.e., 48 of the top 50; Supplementary Table 1) of the SNVs that we infer to be most beneficial for SARS-CoV-2 transmission. Our approach is based on a branching process epidemiological model of viral transmission. This is distinct from "black box" deep learning methods (including large language models) that have been proposed to address related but distinct problems, such as characterizing antigenic evolution and antibody escape dynamics[64,65]. Other groups have taken complementary approaches to inferring differences in SARS-CoV-2 transmission that include individual SNV effects, incorporating different phenotypic effects[66] or transmission models[60].

The epidemiological model that we have introduced has limitations. We assumed a fairly short generation time, which is appropriate for a virus such as SARS-CoV-2. A different approach would be needed to consider the spread of viruses where many transmission events are from long-term infections, such as HIV. We also assume that SNVs contribute additively to fitness and that selection coefficients are constant in time. Our model does not delineate intrinsic (e.g., functional) effects of SNVs on transmission from selection advantages due to immune escape; though for many of the SNVs inferred most strongly to affect the transmission, there is independent experimental evidence to suggest that each (or both) of these factors are important (Supplementary Table 1). In principle, selection for immune escape is likely to be time-varying, as the buildup of population immunity reduces the selective advantage of escape mutations over time[67]. Simulations show that if selection is time-varying, the constant selection coefficients that we infer reflect averages of time-varying selection over the time that the variant was observed (Supplementary Fig. 8). Epistasis could also lead to over- or under-estimation of selection coefficients for specific SNVs, but total contributions to transmission from multiple SNVs are typically estimated accurately (Supplementary Fig. 9). We have also assumed that serial intervals are constant in time, but variants may differ in the typical time between infections[68] which could influence relative growth rates. Differences in antigenicity could also generate fitness differences that are intransitive and which depend on immune history. A model that explicitly incorporates antigenicity would be needed to account for this effect. Finally, we note that no model based solely on dynamics, including ours, could distinguish the independent effects of different SNVs that exclusively appear together on the same genetic background.

Our ability to rapidly identify new, high-growth variants is naturally limited by the public availability of sequence data. Time lags between when sequencing is performed and when sequences are uploaded, in particular, can lead to delays. As shown in Fig. 4, filtering sequences by collection date rather than submission date typically leads to much faster detection of variant growth. The disparity is especially large for Omicron: sequence data from samples collected by mid-October 2021 already shows a substantial transmission advantage for this variant. In Great Britain, early Alpha sequences were significantly more likely to have short delays between collection and submission, causing Alpha sequences to be over-represented in early data and closing the gap between selection estimates. Even in this unusual case, however, earlier reporting substantially reduces noise. Thus, reducing the time between when sequencing is performed and when sequence data is publicly shared could facilitate the detection of new variants with increased transmission and help prepare for growing outbreaks.

Our focus on quantifying the effects of individual SNVs on viral transmission also mitigates some data limitations. Even in cases where sequence data for a novel variant is limited, emerging variants could be identified for further attention based on the presence of previously observed SNVs. For example, Alpha, Delta, and Omicron (BA.1) would have had estimated selection coefficients of $\hat{w} = 18\%$, 17%, and 66%, respectively (relative to the WIV04 reference sequence), immediately prior to their first observations in sequence data. More generally, as shown in Supplementary Fig. 7, for multiple major variants there is evidence that some of their variant-defining SNVs substantially increase transmission prior to the wide circulation of those variants among humans.

While our study has focused on SARS-CoV-2, the epidemiological model that we have developed is very general. The same methodology could be applied to study the transmission of other pathogens such as influenza. Combined with thorough genomic surveillance data, our model provides a powerful method for rapidly identifying more transmissible viral lineages and quantifying the contributions of individual SNVs to changes in transmission.

## Methods

### Epidemiological model

We use a discrete-time branching process to model the spread of infection. Individuals can be infected by any one of $M$ viral variants, which are represented by genetic sequences $\boldsymbol{g} = \{g_1, g_2, ..., g_L\}$ of length $L$. For simplicity, we will first assume that alleles at each site $i$ in the genetic sequence for variant $a$ are either equal to the "wild-type" or reference ($g_i^a = 0$) or mutants ($g_i^a = 1$). Later we will relax this assumption to consider genetic sequences with five possible states at each site (four nucleotides or a gap). We call $n_a(t_m)$ the number of individuals infected by variant $a$ at time $t_m$. To allow for super-spreading, the number of newly infected individuals at time $t_{m+1}$ follows a negative

binomial distribution[12–17], $P(n_a(t_{m+1})|n_a(t_m), k, R_a) = P_{NB}(r, p)$, where $r = n_a k$, $p = k/(k + R_a)$, and $R_a = R(1 + w_a)$. Here $r$ and $p$ are the negative binomial distribution parameters, $k$ is the dispersion, $R$ is the effective reproductive number of the reference variant, and $w_a$ encodes the variant dependence of the infectivity. The parameters $n$, $k$, and $R$ can be time-varying. For instance, a time-varying $R$ can represent a change in the number of susceptible and recovered individuals as well as the effects of public health interventions or changes in behavior that modify viral transmission.

Defining the frequency of variant $a$ as $y_a = n_a/\sum_b n_b$, the probability that the frequency vector is $\boldsymbol{y}(t_{m+1}) = \{y_1(t_{m+1}), y_2(t_{m+1}), \ldots\}$ given the initial frequency vector $\boldsymbol{y}(t_0)$, is

$$P((\boldsymbol{y}(t_m))_{m=1}^T|\boldsymbol{y}(t_0)) = \prod_{m=0}^{T-1} P(\boldsymbol{y}(t_{m+1})|\boldsymbol{y}(t_m)). \tag{2}$$

### Derivation of the estimator

Because Eq. (2) is difficult to work with directly, we follow the approach of ref. 18. We introduce a "diffusion approximation" where we assume that the total number of infected individuals is large and the effects of mutations on transmission are small. Similar approximations have been widely used in population genetics[69–71]. Under these assumptions, the probability distribution for the variant frequencies satisfies a Fokker–Planck equation with terms derived from the first and second moments of the frequency changes $y_a(t_{m+1}) - y_a(t_m)$ under the negative binomial distributions above.

However, the genotype space is high-dimensional (dimension $2^L$, with either a mutant or wild-type allele at each site) and undersampled, making inference of selection for genotypes extremely challenging. To simplify the inference problem, we assume that selection is additive, so the total selection coefficient $w_a$ for variant $a$ is the sum of selection coefficients $s_i$ for mutant alleles at each site $i$:

$$w_a = \sum_{i=1}^L g_i^a s_i. \tag{3}$$

We can then derive a Fokker–Planck expression for the dynamics of mutant allele frequencies

$$x_i = \sum_{a=1}^M g_i^a y_a. \tag{4}$$

At the allele level, the Fokker–Planck equation has a drift vector given by

$$d_i(\boldsymbol{x}) = x_i(1 - x_i)s_i + \sum_{j=1, j\neq i}^L (x_{ij} - x_i x_j)s_j, \tag{5}$$

and a diffusion matrix

$$C_{ij} = \left(\frac{1}{k} + \frac{1}{R}\right) \times \begin{cases} x_{ij} - x_i x_j & i \neq j \\ x_i(1 - x_i) & i = j \end{cases}, \tag{6}$$

where $x_{ij}$ is the frequency of infected individuals that have mutant alleles at both site $i$ and site $j$ at time $t$. In deriving Eq. (5) we have assumed that the selection coefficients satisfy $s_i \ll 1$ such that $w_a \ll 1$. Despite this technical assumption, our simulations demonstrate that selection can be accurately inferred even when selection is strong (Supplementary Fig. 10). The drift vector describes the expected change in allele frequencies over time. Eq. (5) consists of two terms. The first describes the expected change in the frequency of allele $i$ due to selection at that site. The second term accounts for linkage, that is, it

quantifies how the genetic background alters the expected frequency change of an allele.

The Fokker–Planck equation can then be used to derive a path integral, which gives the probability of an entire evolutionary history or "path" (i.e., frequencies of genetic variants over time, $\boldsymbol{x}(t_m)_{m=1}^T$). In Supplementary Information, we derive the path integral expression following a similar approach to the one described in ref. 18. The path integral is

$$P\left((\boldsymbol{x}(t_m))_{m=1}^T|\boldsymbol{x}(t_0), \boldsymbol{s}, n\right) \approx \left(\prod_{m=0}^{T-1} \frac{1}{\sqrt{\det C}} \left(\frac{n}{2\pi\Delta t_m}\right)^{L/2} \prod_{i=1}^L dx_i(t_{m+1})\right) \exp\left(-\frac{n}{2}S\left((\boldsymbol{x}(t_m)_{m=0}^T\right)\right), \tag{7}$$

where

$$S\left((\boldsymbol{x}(t_m)_{m=0}^T\right) = \sum_{m=0}^{T-1} \left[\frac{\boldsymbol{x}(t_{m+1}) - \boldsymbol{x}(t_m)}{\Delta t_m} - \boldsymbol{d}(\boldsymbol{x}(t_m))\right] C^{-1}(\boldsymbol{x}(t_m)) \left[\frac{\boldsymbol{x}(t_{m+1}) - \boldsymbol{x}(t_m)}{\Delta t_m} - \boldsymbol{d}(\boldsymbol{x}(t_m))\right]. \tag{8}$$

Here $n = \sum_{a=1}^M n_a$ is the total number of individuals infected by all variants and $\Delta t_m = t_{m+1} - t_m$. The path integral in Eq. (7) has a form that is similar to the one obtained in ref. 18. The path integral quantifies the probability density for paths of mutant allele frequencies in the evolutionary history of the pathogen. We can then use Bayesian inference to find the maximum *a posteriori* estimate for the selection coefficients given the frequencies, the infected population size, the parameters $R$ and $k$. The posterior probability of the selection coefficients is

$$P\left(\boldsymbol{s}|(\boldsymbol{x}(t_m), n)_{m=0}^T\right) \propto P\left((\boldsymbol{x}(t_m))_{m=1}^T|\boldsymbol{x}(t_0), \boldsymbol{s}, n\right) \times P_{\text{Prior}}(\boldsymbol{s}), \tag{9}$$

where $P\left((\boldsymbol{x}(t_m))_{m=1}^T|\boldsymbol{x}(t_0), \boldsymbol{s}, n\right)$ is the probability of a path given by Eq. (7) and the $P_{\text{Prior}}(\boldsymbol{s})$ is a Gaussian prior probability for the selection coefficients with zero mean and covariance matrix $\sigma^2 I$. Here, $I$ is the identity matrix and $\sigma^2$ is the variance of the prior. We call the precision $\gamma = 1/\sigma^2$. In Supplementary Information, we show that the selection coefficients that maximize Eq. (9) are

$$\hat{\boldsymbol{s}} = \left[\gamma I + \sum_{m=0}^{T-1} \frac{nk^2R^2}{(k+R)^2}\Delta t_m C(t_m)\right]^{-1} \left[\sum_{m=0}^{T-1} \frac{nkR}{k+R}\left(\boldsymbol{x}(t_{m+1}) - \boldsymbol{x}(t_m)\right)\right], \tag{10}$$

where the parameters $k$, $R$, and $n$ are implicit functions of $t$.

There are two interesting limiting forms of the estimator. First, we define the new matrix $\bar{C}$ whose entries are

$$\bar{C}_{ij} = \begin{cases} x_{ij}(t_m) - x_i(t_m)x_j(t_m) & i \neq j \\ x_i(t_m)(1 - x_i(t_m)) & i = j \end{cases}. \tag{11}$$

In the limit that $k \to \infty$, the negative binomial distribution for new infections becomes a Poisson distribution with rate $\lambda = R$. In this special case, the model is equivalent to the Wright–Fisher model from population genetics. The estimator reduces to

$$\hat{\boldsymbol{s}} = \left[\gamma I + \sum_{m=0}^{T-1} nR\bar{C}\right]^{-1} \left[\sum_{m=0}^{T-1} nR(\boldsymbol{x}(t_{m+1}) - \boldsymbol{x}(t_m))\right]. \tag{12}$$

The opposite limit $k \to 0$ corresponds to a distribution for new infections with extremely heavy tails, i.e., one where super-spreading is dominant. In this case, the drift in Eq. (5), which quantifies expected frequency changes due to selection, is unchanged. However, the diffusion matrix, which encodes linkage as well as the changes in frequency that are due to the stochastic nature of infection transmission, diverges. In this case, diffusion dominates the process entirely.

## Simplifying the estimator and robustness to incomplete knowledge of time-varying parameters

While our model has the ability to account for the time dependence of parameters appearing in Eq. (10), such as the infected population size $n$, the dispersion $k$, and the mean reproductive number $R$, these can be challenging to reliably estimate from data. However, we generally do not require full knowledge of these time-dependent parameters to accurately estimate selection.

In fact, due to finite sampling noise, estimates of selection produced by assuming constant (and incorrect) parameters are more accurate than estimates that use the true time-varying parameters (Supplementary Fig. 11). The naive estimator in Eq. (10) implies that time points or regions with larger $R$, $n$, or $k$ should be weighted more heavily in the estimate. However, frequency information is always inaccurate due to noise from finite sampling, so weighing some time points or regions significantly more than others based on the parameters alone means that undue weight is given to the uncertain information available from these times and regions.

For this reason, we assume parameters that are spatially and temporally constant in all of the following analyses, as discussed below. This allows the estimator to be simplified substantially. If we assume constant parameters and scale the regularization $\gamma$ by $nkR/(k+R)$ in the numerator in Eq. (10), the parameter dependence in the numerator and the denominator is identical and cancels out (due to the additional factor of $(k+R)/kR$ in the definition of the covariance matrix). With the same definition of the matrix $\bar{C}$ as above, and additionally defining $\bar{C}_{int} = \sum_{m=0}^{T-1} \Delta t_m \bar{C}$ and $\gamma' = \gamma nkR/(k+R)$, the simplified estimator is given by

$$\hat{s} = \left[\gamma' I + \bar{C}_{int}\right]^{-1} \left[\boldsymbol{x}(t_T) - \boldsymbol{x}(t_0)\right]. \tag{13}$$

This form of the estimator is similar to the estimator for selection coefficients in the Wright–Fisher model[18], except that it omits contributions from the mutation term because the mutation rate for SARS-CoV-2 is small. Practically, Eq. (13) has significant advantages over Eq. (10). The most important is that the difficult-to-estimate parameters $k$ and $n$ are no longer required. In addition, $R$ does not need to be estimated. For methods of inferring these parameters as well as discussions about the difficulty of inferring them, see refs. 72–81.

## Extension to multiple regions and multiple SNVs at each site

The model can easily incorporate data from multiple regions or outbreaks at different times. If the probability of the evolutionary path in each region is independent, which is the case if there is no travel between regions, then the probability of all of the evolutionary paths in all of the regions is simply the product of the probabilities of the paths in each region, given by Eq. (7). Bayesian inference can be applied in the same way as before, resulting in the estimator

$$\hat{s} = \left[\gamma' I + \sum_{r=1}^{Q} \bar{C}_{r,int}\right]^{-1} \left[\sum_{r=1}^{Q} \boldsymbol{x}_r(t_{r,T_r}) - \boldsymbol{x}_r(t_{r,0})\right], \tag{14}$$

where $Q$ is the number of regions, $t_r$ is the time in region $r$, $t_{r,T_r}$ is the final time in region $r$, $t_{r,0}$ is the initial time in region $r$, $\boldsymbol{x}_r$ is the frequency in region $r$, and $\bar{C}_{r,int}$ is the scaled integrated covariance matrix in

region $r$ given by integrating Eq. (11) over time. The estimator can further be extended to allow for multiple different nucleotides at each site by simply letting each different nucleotide have its own entry in the frequency vector $x_i$. If there are $J$ mutations at each site this results in a frequency vector of length $LJ$, and a covariance matrix of size $LJ \times LJ$. By convention, reference sequence alleles have selection coefficients of zero, so the mutant allele selection coefficients at each site are normalized by subtracting the inferred coefficient for the reference allele.

## Branching process simulations

We implemented the superspreading branching process for the number of infected individuals in Python. We used a negative binomial distribution for the number of secondary infections caused by a group of individuals infected with the same pathogen variant. To test how finite sampling affects model estimates, we sampled $n_s$ genomes per time point to use for analysis. We computed the single and double mutant frequencies, $x_i$ and $x_{ij}$, respectively, from the sampled sequences and estimated the selection coefficients from these using Eq. (1), possibly extended to account for multiple outbreaks or multiple alleles at each locus as described above. For the analysis of how finite sampling affects estimates, shown in Supplementary Fig. 11, we use the full version of the estimator given by Eq. (10). For all other simulations, we assume that the parameters $n$, $k$, and $R$ are not known for inference and so we use the simplified estimator in Eq. (14) for inferring selection.

## Regions and time-series for SARS-CoV-2 analysis

We used sequence alignments and metadata downloaded from GISAID (ref. 19) on January 26, 2024, which includes more than 7.4 million sequences. One potential caution in interpreting this data is that not all sequences in the database will have been generated from unbiased surveillance efforts.

Ideally, we would like to divide this data into the smallest separate areas that have outbreaks that are largely independent of those in the surrounding regions, so as to avoid biases due to travel between regions or unequal sampling in different locations. However, this needs to be balanced with the limitations of the data, since regions with poor sampling could contribute more noise than signal. We, therefore, divided data into the smallest regions available in the metadata that are still large enough such that infections resulting from travel outside of the region are likely to be far less frequent than transmission within the region. This results in the inclusion of mostly separate countries in Europe states in North America, and a combination of countries and states in South America and Asia—dependent upon the size of the location. Two exceptions to this are that we separate northern and southern California due to the geographical separation of population centers, and we separate Northern Ireland from the rest of the United Kingdom due to its geographical isolation.

To minimize the effects of sampling noise, we chose regions and time-series within these regions based on the following criteria:
- In any period of 5 days within the time-series there are at least 20 total samples.
- The number of days in the time-series is >20.
- The number of new infections per day is at least 100.

The last criterion ensures that there are enough infected individuals that transmission is not driven overwhelmingly by stochasticity. We assessed the number of newly infected individuals by using the estimates provided by the Institute of Health Metrics and Evaluations[82]. Since the dates provided in their estimates correspond to dates when individuals were infected, and dates in the GISAID sequence data correspond to dates when individuals were sequenced, we shifted the

dates in the IHME data 5 days forward to roughly compensate for delays between infection and sequencing. We then eliminated days on which the estimated number of new infections was smaller than 100.

Our results are robust to reasonable variation in these parameters. Comparing the number of locations used and the sample sizes shown in Supplementary Fig. 12 in the data to those used in the simulations shown in Supplementary Fig. 1, we expect our inference to accurately distinguish beneficial, deleterious, and neutral SNVs from one another.

### Data processing

We perform a number of preprocessing steps to ensure data quality. We first eliminated incomplete sequences with gaps or ambiguous nucleotides at more than 1% of the genome. We then removed sites from our analysis where gaps are observed at >95% frequency, since these sites may represent very rare insertions or sequencing errors. We also removed sites in noncoding regions of the SARS-CoV-2 genome and ones where all observed SNVs are synonymous. We imputed gaps that are not associated with known variants and ambiguous nucleotides with the nucleotide at the same site that occurs most frequently in other sequences from the same region.

For the remaining sites, in each region we excluded rare SNVs whose frequency is not larger than 1% for at least 5 consecutive days. These sites, if included, are almost always inferred to have extremely small selection coefficients. Furthermore, since their frequencies are so small, their covariance with other sites is also small and is, therefore, unlikely to have a large effect on inference. We verified that different reasonable values for these cutoffs result in essentially identical selection coefficients (Supplementary Fig. 13).

### Calculating frequency changes and covariances

To increase robustness to finite sampling in time, we integrated terms in Eq. (10) and other time-dependent equations over time by assuming that frequencies are piecewise linear, rather than summing contributions from each time point[18]. This results in diagonal terms of the integrated covariance given by

$$\sum_{m=0}^{T-1} \Delta t_m \left[ \frac{(3 - 2x_i(t_{m+1}))(x_i(t_m) + x_i(t_{m+1}))}{6} - \frac{x_i^2(t_m)}{3} \right], \quad (15)$$

and off-diagonal elements

$$\sum_{m=0}^{T-1} \Delta t_m \left[ \frac{x_{ij}(t_m) + x_{ij}(t_{m+1})}{2} - \frac{x_i(t_m)x_j(t_m) + x_i(t_{m+1})x_j(t_{m+1})}{3} - \frac{x_i(t_m)x_j(t_{m+1}) + x_i(t_{m+1})x_j(t_m)}{6} \right]. \quad (16)$$

For obtaining reliable estimates of the changes in SNV frequencies (the term $x(t_T) - x(t_0)$ in Eq. (13)), it is important to have enough sequences to avoid large errors due to finite sampling. On the other hand, if a large number of days are used at the end or the start of the time-series to calculate the frequencies, then the frequency changes are likely underestimates. To balance these competing issues, we calculated $x(t_T)$ as the frequencies in the window of the final 15 days and $x(t_0)$ as the frequencies in the window of the first 15 days for each time-series and region with poor sampling. This smoothing is necessary especially in regions where sampling is sparse, where the number of genomes sampled on a particular day may be as small as 1 or 2. If there are at least 200 sampled sequences in a period of less than 15 days at the start or the end of the time-series, then the window size was taken as the smallest number of days in which there was a total of at least 200 sequences. We confirmed that our results are robust to reasonable changes in this window size of 15 days (Supplementary Fig. 13). We also normalized time in units of serial intervals or "generations" by dividing the integrated covariance matrix by 5, following results that the serial interval for SARS-CoV-2 is roughly 5 days[83–85]. This allows us to convert from units of time in days to generations, as in Eq. (13).

### Calculating selection coefficients

After the above preprocessing there remain 1320 SNVs observed at a frequency above 1% for at least 5 consecutive days in at least one region and observed at least 5 times. We assume constant values for $R$, $n$, and $k$ in all regions, and use Eq. (14) to estimate selection. When $R$, $n$, and $k$ are constant, these terms can be effectively absorbed into the regularization $\gamma'$.

We normalize selection coefficients such that the nucleotide for the WIV04 reference sequence at each site has a selection coefficient of 0. To do this, we subtract the selection coefficient for the reference nucleotide from the inferred coefficient for each other allele at that site after all selection coefficients have been computed.

We used these estimates for the selection coefficients for non-synonymous SNVs to estimate the corresponding selection coefficients for amino acid substitutions (Supplementary Table 1). If there were multiple SNVs in a codon that result in the same amino acid variant but are not strongly linked to one another, then the selection coefficient for the amino acid was calculated as the largest (in absolute value) of the SNVs. If there were multiple SNVs in the same codon that yield the same amino acid and these SNVs are strongly linked to one another, then the selection coefficient for the mutant amino acid was calculated as the sum of the selection coefficients for the SNVs. Our reasoning behind this choice is that selection coefficients that are extremely close to zero are mostly for alternative nucleotides that are observed very infrequently in the data, and so the inferred coefficients for these nucleotides are unlikely to reflect the typical effects of a given mutation.

We calculated selection coefficients for major variants by summing the individual nucleotide SNVs that define the variant, which follows from our assumption of additive fitness. The SNVs for major named variants such as Alpha and Delta were identified according to the mutations provided by https://covariants.org. Results of this analysis are shown in Figs. 2, 3, Supplementary Figs. 3–5 and 14, 15 and Supplementary Table 1. Supplementary Figs. 14, 15 quantify uncertainty in the inferred selection coefficients, based on both theoretical uncertainty in the selection coefficient estimator and finite sampling noise. For a detailed discussion, see Supplementary Information.

### Computational complexity

Here we briefly discuss the computational complexity of our method. The steps in our data processing are:

- Clean the data (eliminate sequences with large numbers of Ns or gaps, etc.).
- Separate the data by time and region.
- Identify SNVs observed above the minimum frequency threshold.
- Compute SNV covariance matrices/changes in SNV frequencies in each region and integrate them over time.
- Infer the selection coefficients, which involves inverting the total integrated SNV covariance matrix.

Let $L$ be the length of the SARS-CoV-2 sequence (roughly $3 \times 10^4$ bps) and let $M$ be the total number of sequences (roughly $10^7$, including data taken up until January 26th, 2024). Then, steps 1 and 2 involve computations that scale as $\mathcal{O}(M)$. Step 3 is $\mathcal{O}(ML)$. This step also introduces a new parameter relevant to the scaling of the problem, which is the fraction of SNVs that are observed at high enough frequencies to be included in our analysis. Let us call this fraction $p$, which is roughly 0.35 with our current settings. Naively, step 4 then involves a computation that scales like $\mathcal{O}(M(pL)^2)$. However, the calculation of the covariance can easily be parallelized across regions. In each individual region, the fraction of SNVs that are observed at high enough frequencies to be included is a different parameter $q$ and the number of sequences in the region is a parameter $M_r$. The largest $q$ that we find in the regions analyzed is around 0.05. For $N_r$ separate regions (149 in our analysis), step 4 then involves $N_r$ parallel computations that

scale-like $\mathcal{O}(M_r(qL)^2)$. Due to the matrix inversion, step 5 requires $\mathcal{O}((pL)^3)$ computations to complete.

## Choice of regularization

In principle, the regularization strength $\gamma'$ is related to the width of the prior distribution for SNV selection coefficients. The regularization strength also plays a role in reducing noise in selection coefficient estimates due to the finite sampling of viral sequences. This is especially important for SNVs that are observed only briefly in data, as they will have small integrated variances in the "denominator" of Eq. (10). Larger values of the regularization more strongly suppress noise, but they also shrink inferred selection coefficients towards zero.

We use a regularization strength of $\gamma' = 40$. For much smaller values of $\gamma'$, selection coefficient estimates are unstable due to sampling noise. However, inferred selection coefficients stabilize and become insensitive to the precise value of $\gamma'$ for $\gamma' \gtrsim 10$ (Supplementary Fig. 13). Larger values of $\gamma'$ will result in selection coefficients with smaller absolute values, but for large enough $\gamma'$ the rank ordering of inferred selection coefficients is highly reliable. In summary, the coefficients that appear to be the most beneficial or deleterious remain this way regardless of reasonable choices for $\gamma'$, though their precise values scale with the regularization strength.

## Identification of HG SNVs

To estimate how quickly we can detect a transmission advantage for a new SNV or variant, and to explore the sensitivity of this detection, we inferred selection coefficients for all variants $\hat{w}$ (including SNVs and collections of SNVs that are strongly linked to one another), for every day in every region separately. To determine sets of strongly linked SNVs, we considered the following statistics. If the number of genomes with an SNV at site $i$ is called $h_i$ and the number of genomes with SNVs at both site $i$ and site $j$ is $h_{ij}$, then we say that two sites $i$ and $j$ are strongly linked if $h_{ij}/h_i$ and $h_{ij}/h_j$ are both >80%.

To form sets of strongly linked SNVs, we combined all pairs of strongly linked SNVs that share SNVs in common. For example, if SNV $i$ is strongly linked with SNV $j$, and SNV $j$ is strongly linked with SNV $k$, then $\{i, j, k\}$ forms one set of strongly linked SNVs. With the frequency cutoff that we have used for the definition of strongly linked SNVs (80%), the great majority of SNVs in each set of strongly linked SNVs are strongly linked to all other SNVs in the same set. We computed selection coefficients for sets of strongly linked SNVs by summing the contributions from individual SNVs.

Data was trimmed by submission date such that the selection coefficients for a specific day were calculated using only sequences that were submitted to GISAID on or before that day. We then progressively step through time in each region, adding newly submitted sequences and reanalyzing the data again. At each time point in every region, groups of strongly linked SNVs are recalculated using the method described above, and selection coefficients for the collections are computed again. To compare the HG SNVs with well-studied major SARS-CoV-2 variants, which are widely understood to have a significant transmission advantage relative to ancestral SARS-CoV-2, we performed this analysis using data from the beginning of the pandemic through June 2022.

As described in the main text, we suspect that collections of SNVs with large inferred selection coefficients are much more likely to exhibit real advantages in transmission. Therefore, we used a classification scheme where variants with selection coefficients $\hat{w} > \theta$ for some cutoff $\theta$ are classified as "high growth (HG)" variants. At each time step, we removed any SNVs that were classified as HG from all future analyses in that region. In this way, any SNV can only contribute to the detection of a single variant in a region (e.g., for a mutation that belongs to both Alpha and Omicron, if the mutation was labeled as HG during the rise of Alpha in a given region, then that mutation will not be

considered when analyzing later Omicron sequences in the same region).

After a mutation is detected in a region, we also remove all other nucleotide mutations at that site from future analysis in the region. The reason for this is the following. The choice of a normal prior distribution on the selection coefficients enforces that the sum of the selection coefficients for a specific site is zero. We then re-normalize the selection coefficients so that the selection coefficient for the WIV04 reference nucleotide is set to zero. This is done by subtracting its value from the selection coefficients for all other nucleotides at that site, as described above. In the ordinary situation where only two different nucleotides are observed at a site, this normalization procedure results in the apparent inflation of selection coefficients for unobserved nucleotides at the same site. If one of these other nucleotides is later observed at a low frequency, this could result in an incorrect detection. For this reason, we remove all nucleotides at the same site from consideration in a region after any single nucleotide has been detected.

We performed inference for the detection of HG variants across each region individually, as the same new variant is unlikely to first appear at identical times in multiple regions. This limits the strength of statistical information to infer selection because information is not aggregated across regions. For this reason, we used a lower regularization of $\gamma' = 10$ for regional analysis to prevent the strong suppression of inferred selection coefficients. Tuning the threshold of detection $\theta$ allows one to adjust the tradeoff between noise, which may lead to false positives and detection speed. Results of this analysis are presented in Supplementary Figs. 6, 7. The analysis shown in Fig. 4 uses an analogous approach where selection coefficients were computed over time for Alpha, Delta, and Omicron (BA.1) SNVs in specific regions, but without the additional step of classifying SNVs as HG.

To succinctly visualize HG SNVs linked with major variants (Supplementary Fig. 7), we grouped the regions into 7 broad categories, allowing for clearer trend analysis. For each major variant within these broad regions, we identified HG groups with associated mutations and plotted the cumulative fraction of variant-defining mutations over time. Data regarding variant-defining mutations was sourced from https://covariants.org.

## Reporting summary

Further information on research design is available in the Nature Portfolio Reporting Summary linked to this article.

## Data availability

Original SARS-CoV-2 sequence data and metadata used in this project are available through GISAID. A full list of originating and submitting laboratories for the sequences used in our analysis can be found at https://www.gisaid.org using the EPI-SET-ID: EPI_SET_240815xt. Other processed data and data from simulations are available at the GitHub repository[86], https://github.com/bartonlab/paper-SARS-CoV-2-inference.

## Code availability

Computer code and scripts that we have used in our analysis are available in the GitHub repository[86], https://github.com/bartonlab/paper-SARS-CoV-2-inference. This repository also contains Jupyter notebooks that can be run to reproduce our results, based on the data described above. Code for analysis is written in Python (version 3.11.11), including packages numpy (version 1.26.4), scipy (version 1.14.1), pandas (version 2.2.2), and matplotlib (version 3.9.3).

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

## Acknowledgements

We gratefully acknowledge all data contributors, i.e., the Authors and their Originating laboratories responsible for obtaining the specimens, and their Submitting laboratories for generating the genetic sequence and metadata and sharing via the GISAID Initiative, on which this research is based. The work of B.L., E.F., and J.P.B. reported in this publication was supported by the National Institute of General Medical Sciences of the National Institutes of Health under Award Number R35GM138233. The work of A.A.Q., M.S.S., and M.R.M. was supported by the Research Grants Council of the Hong Kong Special Administrative Region, China under Project No. T11-705/21-N; A.A.Q. and M.S.S. were also supported under Project No. 16204121. M.R.M. is the recipient of an Australian Research Council Future Fellowship (Project No. FT200100928) funded by the Australian Government.

## Author contributions

All authors contributed to methods development, data analysis, interpretation of results, and writing the paper. B.L. and J.P.B. led theoretical analyses. M.S.S. and B.L. led simulations. A.A.Q. led validation of SARS-CoV-2 inference results. J.P.B. conceptualized the project. J.P.B. and M.R.M. supervised the overall project.

## Competing interests

The authors declare no competing interests.
