## [Transparent Peer Review file · Nature Communications]

Inferring effects of mutations on SARS-CoV-2 transmission from genomic surveillance data

Corresponding Author: Professor John Barton

Version 0:

Reviewer comments:

Reviewer #3

(Remarks to the Author)

This paper has been through many rounds of review, and there is a lot of good material here that I believe merits publishing in this journal.

I still don't think that the authors have properly grappled with requests, not just from me, to evaluate the sensitivity and specificity of their method in a way that is comprehensible to their target audience. That is a shame, because dealing with that properly would make this a stronger paper. But I don't think there is anything to be gained from further discussion of this point. I appreciate that the authors have removed most references to the method functioning as an "early warning system", so the paper is no longer making a claim that it doesn't properly support.

I still have an issue with the sentence:

"Our model incorporates superspreading by drawing the number of secondary infections caused by an infected individual from a negative binomial distribution with mean R , referred to as the effective reproduction number, and dispersion parameter k (refs. 12–17)."

I think I have been clear about this. The model incorporates superspreading, sure, but none of the results being shown here are impacted by superspreading because the authors conclude that the effect of this would be small and difficult to estimate.

As I said last time: "The authors have only shown (1) that superspreading is not likely to be important for estimating selection coefficients and (2) that it would be hard to properly account for superspreading anyway, since we do not have the right data to do this. I do not find these findings to be surprising. And crucially, the actual results presented in the paper (estimates of fitness effects, etc) do not account for superspreading, since the authors have decided to (justifiably) ignore the impact of superspreading after showing theoretically that there is little point in going to the effort of doing so."

The sentence from the paper that I quoted above would be acceptable if something like the following amendments were made. (I'm putting the changes in capitals so they stand out, not to be shouty):

"Our model CAN INCORPORATE superspreading by drawing the number of secondary infections caused by an infected individual from a negative binomial distribution with mean R , referred to as the effective reproduction number, and dispersion parameter k (refs. 12–17). HOWEVER, WE SHOW IN THE SUPPLEMENT THAT [reason why the superspreading parameter is dropped from the formulation], AND ACCORDINGLY THE DEGREE OF SUPERSPREADING DOES NOT AFFECT THE RESULTS SHOWN HERE."

Or something similar.

(Remarks on code availability)

Reviewer #4

(Remarks to the Author)

This is a manuscript with significant revision and adaptation in response to two previous rounds of review. There is more focus on the question of estimating transmissibility and less emphasis on aspects of forecasting. There are however some questions on definitions and wording as well as a couple of technical comments.

1. The manuscript maintains ambiguity in choice of terms. The abstract now places emphasis on “mutation” vs variants (without definition of mutation), while most of the text sticks to the terms SNV and variants/VOCs.
2. The refer to SNV/variants associated with growth (transmissibility) but also refer loosely to SNV/variants associated with “selection” (eg., To test our ability to reliably infer selection”). This again would benefit from a clear choice of words and concepts across the text.
3. The following text is important and should show up prominently in the manuscript (eg. Abstract or and of Introduction): “our model uses genomic surveillance data to estimate the effects of different mutations, it does not make predictions about the effects of mutations that have never been observed before.”
4. Fig 3. Is shown to illustrate the evolution in transmissibility. It seems that the rapid increase in “selection coefficient” is just driven by the fact that evolving VOCs are defined by larger numbers of SNVs and that the additive model results on expected jumps in total selection coefficient values. This may defer on actual/biological selection coefficients of the VOC themselves
5. Fig. 4. Panel 4c may need more discussion. The solid line reflects the post-hoc understanding of which SNVs converged to result in Omicron. They are endowed individually with high selection values – and not as stated “sequence data collected by mid-October 2021 already shows a substantial transmission advantage for this variant” – Omicron was not present at low levels at that time. The sharp increase in selection coefficient of the dotted line is the public recognition of Omicron as a true emergent VOC. Actually, this concept was already described in Maher et al (<https://pubmed.ncbi.nlm.nih.gov/35014856/>).

(Remarks on code availability)

Version 1:

Reviewer comments:

Reviewer #4

(Remarks to the Author)

Thanks for the thorough revision.

(Remarks on code availability)

RESPONSE TO REVIEWERS' COMMENTS

We thank the reviewers again for their comments. Below, we respond to the reviewers' comments and describe our modifications to the manuscript. The reviewers' comments are reproduced in **bold** and our modifications or additions are written in blue.

REVIEWER #3

This paper has been through many rounds of review, and there is a lot of good material here that I believe merits publishing in this journal.

I still don't think that the authors have properly grappled with requests, not just from me, to evaluate the sensitivity and specificity of their method in a way that is comprehensible to their target audience. That is a shame, because dealing with that properly would make this a stronger paper. But I don't think there is anything to be gained from further discussion of this point. I appreciate that the authors have removed most references to the method functioning as an "early warning system", so the paper is no longer making a claim that it doesn't properly support.

I still have an issue with the sentence:

"Our model incorporates superspreading by drawing the number of secondary infections caused by an infected individual from a negative binomial distribution with mean R , referred to as the effective reproduction number, and dispersion parameter k (refs. 12–17)."

I think I have been clear about this. The model incorporates superspreading, sure, but none of the results being shown here are impacted by superspreading because the authors conclude that the effect of this would be small and difficult to estimate.

As I said last time: "The authors have only shown (1) that superspreading is not likely to be important for estimating selection coefficients and (2) that it would be hard to properly account for superspreading anyway, since we do not have the right data to do this. I do not find these findings to be surprising. And crucially, the actual results presented in the paper (estimates of fitness effects, etc) do not account for superspreading, since the authors have decided to (justifiably) ignore the impact of superspreading after showing theoretically that there is little point in going to the effort of doing so."

The sentence from the paper that I quoted above would be acceptable if something like the following amendments were made. (I'm putting the changes in capitals so they stand out, not to be shouty):

"Our model CAN INCORPORATE superspreading by drawing the number of secondary infections caused by an infected individual from a negative binomial distribution with mean R , referred to as the effective reproduction number, and dispersion parameter k (refs. 12–17). HOWEVER, WE SHOW IN THE SUPPLEMENT THAT [reason why the

superspreading parameter is dropped from the formulation], AND ACCORDINGLY THE DEGREE OF SUPERSPREADING DOES NOT AFFECT THE RESULTS SHOWN HERE."

Or something similar.

We appreciate the need for clarity in these statements. Building on the reviewer's suggestion above, we propose the following amendment, with changes/additions highlighted in blue:

Our model **draws** the number of secondary infections caused by an infected individual from a negative binomial distribution with mean R , referred to as the effective reproduction number, and dispersion parameter k . **The negative binomial distribution for secondary infections has been used in past work to model superspreading¹²⁻¹⁷, which becomes more prominent when k is small. However, we show in the **Supplementary Information** that the incorporation of detailed information about the current number of infections and degree of superspreading — even if known perfectly — can actually lead to worse inference when data is finitely sampled. Thus, we will ultimately absorb these terms into a regularization parameter, and the degree of superspreading does not influence the inferred transmission effects of SARS-CoV-2 mutations that we will describe below.**

Along the lines suggested above, this statement more clearly delineates the details that are present in the mathematical model of viral transmission and which aspects of the model are ultimately reflected in the selection coefficients that we infer from SARS-CoV-2 data.

REVIEWER #4

This is a manuscript with significant revision and adaptation in response to two previous rounds of review. There is more focus on the question of estimating transmissibility and less emphasis on aspects of forecasting. There are however some questions on definitions and wording as well as a couple of technical comments.

1. The manuscript maintains ambiguity in choice of terms. The abstract now places emphasis on “mutation” vs variants (without definition of mutation), while most of the text sticks to the terms SNV and variants/VOCs.

In the abstract and part of the introduction, we have used the term “mutation” in the generic sense of genetic change. This part of the text is aimed at a broad audience, and as such, we have maintained this generic description in the abstract and the first part of the introduction. To improve clarity in the main text however, in the latter part of the introduction we now include a clear definition of SNVs/mutations/variants, which we apply consistently throughout the rest of the text:

For clarity, we will refer to **all** non-reference nucleotides (including deletions or insertions) as SNVs and viral lineages possessing common sets of SNVs as *variants*. **When we discuss the effects of amino acid substitutions, we will use the term *mutations* to**

distinguish these substitutions from variation at the nucleotide level, as multiple different nucleotide changes can lead to the same change at the level of proteins.

We maintain these different terms because, at different times, we discuss the inferred effects of 1) SNVs, 2) amino acid changes, which may be mediated by one or more SNVs, and 3) sets of SNVs that collectively define SARS-CoV-2 variants, including VOCs such as Alpha, Delta, and so forth.

2. The refer to SNV/variants associated with growth (transmissibility) but also refer loosely to SNV/variants associated with “selection” (eg., To test our ability to reliably infer selection”). This again would benefit from a clear choice of words and concepts across the text.

We have now added an explanation following the definition of the selection coefficients to make the intended parallel with concepts in population genetics more transparent:

In analogy with population genetics, we refer to the w_a and s_i as selection coefficients. We will maintain this analogy throughout this work, associating natural selection or fitness with the relative capacity for viral transmission between hosts.

We have also made additional minor changes throughout the text to help ensure that these concepts are applied consistently.

3. The following text is important and should show up prominently in the manuscript (eg. Abstract or and of Introduction): “our model uses genomic surveillance data to estimate the effects of different mutations, it does not make predictions about the effects of mutations that have never been observed before.”

We have now included a statement to this effect in the Introduction:

As our approach is based on surveillance data, it infers the effects of observed SNVs, rather than predicting the effects of SNVs that have never been observed before.

4. Fig 3. Is shown to illustrate the evolution in transmissibility. It seems that the rapid increase in “selection coefficient” is just driven by the fact that evolving VOCs are defined by larger numbers of SNVs and that the additive model results on expected jumps in total selection coefficient values. This may defer on actual/biological selection coefficients of the VOC themselves

In general, the inferred variant selection coefficients w are not *a priori* proportional to the number of SNVs that define the variants. Our model infers selection coefficients that best explain collective frequency changes, including *correlations* between SNVs. To see this mathematically, see Eq. (3) in **Methods**. For example, a variant with a transmission advantage of $w = 5\%$

relative to co-circulating variants would grow at the same rate over time regardless of the number of nucleotide differences it has relative to the reference sequence. In simulations, **Supplementary Fig. 10a-b** shows an example of inference for two variants with the same selection coefficient w , but which contain different numbers of mutations. Our method infers similar overall selection coefficients w in each case. In SARS-CoV-2 data, **Supplementary Fig. 5** shows that the SNV selection coefficients s that are summed to compute the variant selection coefficients w vary widely in their effects. **Supplementary Fig. 15** shows that indeed statistical evidence exists to separate the effects of individual SNVs, as the great majority of them are not completely correlated in the data. As an additional example, one can consider the Gamma and Delta variants, which arose after Alpha. Even though Gamma and Delta have fewer mutations than Alpha compared to the SARS-CoV-2 reference sequence, they are inferred by our model to have higher rates of transmission (i.e., $W_{\text{Gamma}}, W_{\text{Delta}} > W_{\text{Alpha}}$).

5. Fig. 4. Panel 4c may need more discussion. The solid line reflects the post-hoc understanding of which SNVs converged to result in Omicron. They are endowed individually with high selection values – and not as stated “sequence data collected by mid-October 2021 already shows a substantial transmission advantage for this variant” – Omicron was not present at low levels at that time. The sharp increase in selection coefficient of the dotted line is the public recognition of Omicron as a true emergent VOC. Actually, this concept was already described in Maher et al (<https://pubmed.ncbi.nlm.nih.gov/35014856/>).

To be clear, the solid lines reflect selection coefficient estimates based on actual sequence data in the GISAID repository. Omicron samples had been collected by October 2021. The frequency of Omicron samples in South Africa in GISAID, according to the sample collection date, is shown in the bottom row in **Fig. 4c**. However, since there is a delay between when samples are collected, when sequencing is performed, and when sequence data are uploaded to online repositories, one does not have access to complete information in real time. We agree, then, that these Omicron samples may not have been sequenced by mid-October. Thus, to improve the clarity of this sentence we now write:

...sequence data from samples collected by mid-October 2021 already shows a substantial transmission advantage for this variant.

The selection coefficient that we infer based on sequence data accessible in real time is described by the dashed line. The sharp increase of the dashed line is associated with the release of full-genome Omicron sequences to the GISAID database, which occurred on December 7, 2021. In fact, Omicron had already been declared a VOC by the time this data was released. Our plot also shows only the selection coefficient for *novel* Omicron mutations, not ones that had previously been observed, which appears to be different from the analysis of Omicron by Maher et al. Nonetheless, this (Maher et al.) is a relevant paper describing the analysis of SARS-CoV-2 evolution, which we now cite in the Discussion where we discuss alternate methods.

Sincerely,
John Barton
Associate Professor
Department of Computational and Systems Biology
University of Pittsburgh School of Medicine

Matthew McKay
Professor
Department of Microbiology and Immunology, Peter Doherty Institute for Infection and Immunity,
Department of Electrical and Electronic Engineering, University of Melbourne,
Victorian Infectious Diseases Reference Laboratory, Royal Melbourne Hospital